# Responsible Imputation of User Behavior Surveys via Mask-Aware Transformers

**Aman Shukla** [*]
Resonate Networks, Inc.
Virginia, USA

**Rishabh Kumar**
Resonate Networks, Inc.
Virginia, USA

**Daniel Patrick Scantlebury** [†]
Resonate Networks, Inc.
Virginia, USA

## Abstract

User behavior data collected through surveys is foundational to applications in AdTech, personalization, and consumer intelligence. However, the structured nature of survey fielding governed by routing logic, platform constraints, and user fatigue results in pervasive missingness that is non-random and logic-driven. These gaps hinder the effectiveness of downstream systems that rely on user representations. We present a Transformer-based framework for imputing missing responses in multi-choice behavioral survey data. Our model encodes survey responses as flattened multi-hot vectors with associated binary masks indicating fielded questions. Through column-wise attention and mask-aware supervision, the model learns high-fidelity imputations while honoring routing logic. To enforce plausibility, we apply strict logical enforcement that filters predictions based on domain-aligned consistency rules. Empirically, we evaluate imputation performance under synthetic masking across increasing sparsity levels, demonstrating robust F1 and recall even in highly incomplete settings. Our ablation studies confirm the importance of structured attention and supervision masking. We further conduct a responsible imputation audit, assessing fairness across age, gender, and ethnicity- capturing both model fit and outcome parity. The results reveal stable performance across subgroups, indicating suitability for equitable industrial deployment. Our approach closes a critical gap between modeling sophistication and real-world deployment constraints in survey data pipelines, setting a precedent for responsible and scalable imputation.

## 1 Introduction

Survey-based data collection is a cornerstone of user behavior modeling across a wide range of industries, including AdTech, market research, political polling, and personalized recommendations. Unlike passively logged behavioral data, survey responses offer a direct, interpretable lens into user attitudes, preferences, and intentions; all of which are critical for building robust downstream systems. In large-scale platforms, surveys are routinely deployed to tens of millions of users, capturing granular traits ranging from media habits and product affinity to lifestyle preferences and purchasing intent.

However, despite their interpretability, survey datasets are often riddled with structured missingness. Users are typically shown only a subset of the total questionnaire due to a combination of factors including routing logic, fatigue management, regulatory constraints, and business-specific survey design. For instance, a user's answer to one question may gate whether subsequent questions are shown; others may be shown only in specific regions or demographics. The resulting dataset, once transformed into a machine-readable tabular format, contains a sparse binary vector for each user,

---

[*] Send any correspondence about the work to aman.shukla@resonate.com
[†] Work done during his time at Resonate Networks, Inc.

39th Conference on Neural Information Processing Systems (NeurIPS 2025) Workshop: Reliable ML from Unreliable Data.

where positive entries indicate selected responses and the remaining entries are either implicitly unobserved or explicitly not shown. Importantly, this missingness is highly non-random and logic-driven often violating assumptions made by traditional imputation techniques which assume randomness in the missing data.

This structured sparsity presents a fundamental challenge for modeling. High-dimensional multi-hot vectors, often with thousands of possible response keys, must be interpreted despite having only a small fraction of positive entries per user. Moreover, any system designed to fill in these missing responses or impute behavior must not only generalize effectively but also adhere to logical and regulatory constraints. For example, it would be unacceptable for a model to infer alcohol consumption for a pregnant woman, or to predict mutually exclusive selections in a single-choice question as simultaneously true.

To address these challenges, we propose a Transformer [12]-based architecture specifically tailored for imputation over structured survey data. Unlike standard sequence models, our formulation treats survey responses as unordered feature sets and introduces column-wise attention to capture inter-question relationships. A key feature of our method is masked supervision; the model is trained only on entries for which responses were observed, avoiding the pitfalls of noisy or unverifiable gradients. In addition, we incorporate a causally aligned mechanism to enforce consistency, prevent logically invalid outputs, and align predictions with downstream constraints.

The contributions of this work are twofold. First, we present an imputation framework that is compatible with survey data collection logic, scalable to massive production workloads, and robust to extreme sparsity. We evaluate the model under both natural and synthetic missingness settings, introducing controlled mask perturbations to test generalization. Second, we introduce a responsible audit framework that evaluates fairness and consistency across sensitive user subgroups. Taken together, these contributions lay the foundation for scalable and fair imputation systems that can serve as infrastructure for a range of applications in behavior modeling and personalization that rely on surveys.

## 2 Related Work

Classical statistical approaches such as mean/mode imputation, regression-based filling, and k-nearest neighbors [3] have long been used for dealing with missing values in survey datasets. Among the most commonly used is Multiple Imputation by Chained Equations (MICE) [11], which performs conditional modeling for each variable. It assumes that data is missing at random and that joint distributions can be reliably estimated under that assumption. However, in our setting, where missingness is governed by deterministic survey logic and routing constraints such assumptions are violated, often leading to biased or unreliable imputations. Reference [10] extends the idea by using random forests to iteratively impute missing values. While it improves robustness and is suitable for mixed-type data, it scales poorly to high-dimensional sparse settings and still cannot incorporate logical dependencies or feature hierarchies inherent in survey structures.

Deep learning has introduced more flexible imputation techniques capable of modeling complex dependencies. GAIN [13] applies adversarial training to estimate missing data by treating imputation as a data generation task. While powerful, GAIN requires careful training to stabilize the generator-discriminator dynamics and performs suboptimally in sparse settings with deterministic missingness patterns. HI-VAE [8] combines VAEs with specialized likelihoods for categorical, ordinal, and continuous data. It provides better support for heterogeneous data types but assumes full input during training and does not directly support structured supervision through masking, making it difficult to apply in our context. More recently, VIME [14] explores self and semi-supervised imputation, but it targets dense tabular inputs and does not generalize well to user level logic-based sparsity.

Transformers have seen growing adoption in non-sequential domains, especially for tabular data. TabTransformer [7] embeds categorical variables and applies attention over them in supervised settings. SAINT [9] extends this by applying inter-column attention in a row-wise fashion, and FT-Transformer [6] integrates numerical and categorical features through learned tokenization. However, most of these models assume full input availability at train time and are focused on supervised prediction tasks, not imputation. Moreover, they do not explicitly model missingness or apply masking mechanisms that distinguish observed vs. unobserved fields. SAITS [4], by contrast, applies Transformers to time-series imputation by modeling the temporal and feature-wise relationships

jointly. While SAITS does leverage masking and is tailored for missing data, its assumption of temporal ordering does not hold for our use case involving binary multi-hot survey vectors with no natural sequence. Our model draws on this line of work but retools the attention mechanism for unordered, sparse, high-dimensional survey data, explicitly incorporates structured supervision through masking, and introduces a decoupled causal layer to enforce logic.

There is growing recognition that imputation models can amplify existing biases when errors disproportionately affect protected groups. Reference [5] discuss fairness in algorithmic outcomes and suggest that pre-processing stages, including imputation, warrant scrutiny. Reference [2] conducted a comprehensive study analyzing how various imputation techniques impact fairness in machine learning. Their findings indicate that the choice of imputation method can substantially alter fairness outcomes, emphasizing the need for careful selection of imputation strategies in fairness-critical applications.

## 3    Problem Formulation

In large-scale survey deployments, each user is exposed to a personalized subset of questions based on routing logic, gating conditions, or experimental design. As a result, the final dataset is characterized by structured, non-random missingness. To model this appropriately, we formalize the data and imputation task as follows.

Let $Q = \{q_1, q_2, ..., q_m\}$ be a set of survey questions, where each question $q_i$ is associated with a set of possible response options $O = \{o_1, o_2, ...o_j\}$. In our survey design, depending on the nature of the question, the response type can fall into one of three categories - single, multi and binary select. For single select, the user is allowed to select exactly one response (e.g., "Which streaming platform do you use most?"). For multi-select questions, the user may select one or more responses (e.g., "Which types of media do you consume weekly?") and for binary select, the question is encoded as a presence/absence of a single response key, i.e., a missing selection is interpreted as the complement (e.g., "Do you drink energy drinks?").

Each response option across all questions is assigned an unique key. Let the total number of response keys across all questions be $D$. These keys are flattened into a binary response vector $x_i \in \{0, 1\}^D$ for each user $i$, where each entry corresponds to whether the user selected the associated option. In addition, each user is associated with a binary mask vector $m_i \in \{0, 1\}^D$, where $m_{i,k} = 1$ indicates that the user was shown the question associated with key $k$, and $m_{i,k} = 0$ indicates that the question was not asked. This distinction is crucial, as it indicates

- If $m_{i,k} = 1$ and $x_{i,k} = 1$, the user selected the response.
- If $m_{i,k} = 1$ and $x_{i,k} = 0$, the user saw the response but did not select it.
- If $m_{i,k} = 0$ the user was not shown the question, and $x_{i,k}$ is unknown.

This setup leads to a partially observed multi-hot representation per user, where meaningful supervision is possible only at the positions where $m_{i,k} = 1$. The imputation objective is to predict the full response vector $\hat{x}_i \in \{0, 1\}^D$, estimating the likelihood of each potential response, including those for which $m_{i,k} = 0$. Importantly, the model must learn from observed user behavior to impute missing entries in a logically and contextually consistent manner. The task is framed as a supervised learning problem, where the model learns to predict $x_{i,k}$ only for dimensions where $m_{i,k} = 1$, and generalizes this behavior to unseen portions of $x_{i,k}$ (where $m_{i,k} = 0$) during inference.

## 4    Methodology

Our imputation model is designed to operate over structured survey data represented in a flattened, multi-hot encoding format. Each user is mapped to a high-dimensional binary vector indicating which response keys were selected. Due to survey routing logic, the majority of entries are structurally not observed rather than simply unselected. A depiction of the data collection strategy is shown in Fig 1. The corresponding binary mask indicates which entries were shown to the user. Our architecture processes this sparse binary input using a masked, noise-regularized Transformer framework with a column-wise attention mechanism and causally-aligned output processing.

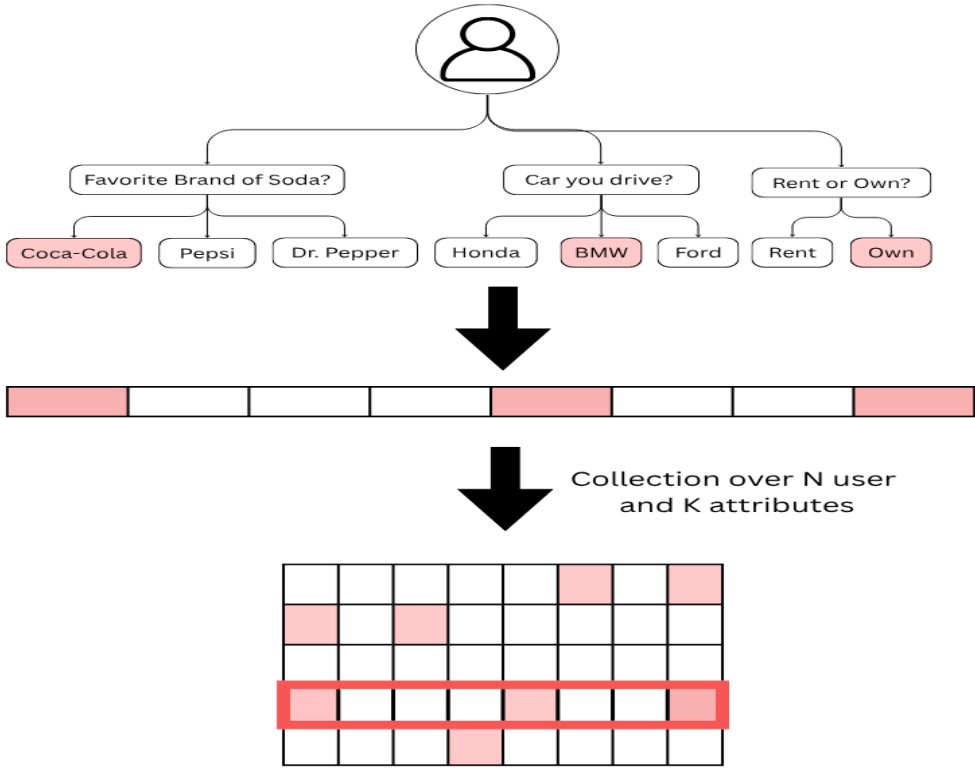

Figure 1: Illustration of the data collection. Users respond to a subset of questions (K) drawn from a larger survey, with selections made from categorical response options. These are flattened into a multi-hot binary vector representing selected and unselected responses. Aggregated across users, this produces a sparse binary matrix of shape $N \times D$ where $N$ are the total users and $D$ are the total number of response keys.

## 4.1 Input Representation and Masking

Let $x \in \{0,1\}^D$ denote the input vector for a given user, where $D$ is the total number of unique response keys in the survey. Each position $x_i = 1$ indicates that the user selected the associated response key and $x_i = 0$ indicates that it was not selected or not shown. To distinguish between these two cases, we introduce a binary mask vector $m \in \{0,1\}^d$, where $m_i = 1$ means the corresponding question was asked and a valid label exists for $x_i$, while $m_i = 0$ denotes unfielded positions. The imputation objective is to predict the values of unobserved entries (where $m_i = 0$), while training is conducted only on known entries (where $m_i = 1$) to ensure validity of supervision.

## 4.2 Embeddings and Self-Attention

Each response key is associated with a unique, learnable embedding vector. We define an embedding table $E \in \mathbf{R}^{D \times e}$, where each row $E_i$ corresponds to a response key and $e$ is the embedding dimension. These embeddings are looked up for all response keys, preserving consistent input length and allowing the model to learn inter-key dependencies even among unobserved entries. The resulting sequence of embeddings is passed into a stack of Transformer encoder blocks, where attention is computed across columns, not temporal or positional indices. This column-wise attention enables the model to capture latent semantic relationships across the entire response space. This workflow is shown in Fig 2.

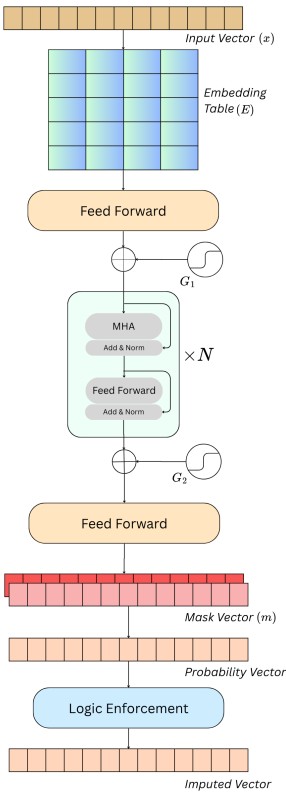

Figure 2: Model architecture. A user's binary input vector $x$ is embedded via a learnable embedding table $E$. Gaussian noise is injected both before and after the transformer block ($G_1, G_2$) to encourage robustness. The output is filtered using the mask vector $m$ to compute loss only on valid entries. A logic enforcement layer applies logical rules to convert probability vectors into final imputed outputs.

## 4.3 Noise Injection

To improve generalization in the presence of extreme sparsity and encourage resilience to input variability, we incorporate Gaussian noise injection at two stages of the model: pre-attention and post-attention. Gaussian noise $G_1$ is added to the embeddings before they enter the transformer block. This corruption simulates uncertainty in the input space and forces attention heads to rely on distributed cues. After the computation, another noise term $G_2$ is added to the attention outputs before they proceed to the prediction head. This second injection helps regularize contextualized representations and smooths the learning signals.

## 4.4 Mask-Aware Supervision

Supervision is applied only in dimensions where valid ground truth is available. During training, we compute the binary cross-entropy (BCE) loss over the positions where $m_i = 1$, ensuring that the model is not penalized for predictions on entries that were never shown to the user. Formally, the loss is given by:

$$\mathcal{L} = \sum_{i=1}^{D} m_i \cdot BCE(y_i, \hat{y}_i) \tag{1}$$

where $y_i \in \{0, 1\}$ is the true label and $\hat{y}_i \in \{0, 1\}$ is the model's prediction. This masked loss encourages focused learning and avoids introducing noise from unverifiable labels. It also aligns tightly with the survey routing logic, as the model is only asked to reconstruct the subset of behavior that was actually observed.

## 4.5 Prediction and Causal Alignment

At inference time, the model produces a probability vector $\hat{y}_i \in [0,1]^D$ via sigmoid activation. To align with downstream business constraints and enforce logical consistency, a logical enforcement layer is applied to the predictions. This logic is a bayesian optimization algorithm that aims to match the predicted distribution with the observed distribution for each response key. For single-select questions, only the highest-scoring response is retained; for multi-select and binary questions, an adaptive threshold is used to determine inclusion. This process is critical to ensure that the imputations remain interpretable and consistent for downstream consumption.

# 5 Experiments and Evaluations

In high-dimensional survey data, where the label space is dominated by the absence of responses, the choice of evaluation metrics must be made with care. While metrics such as accuracy and area under the ROC curve (AUC) are commonly used in binary classification tasks, they can present a misleading picture in sparse multi-hot settings [1] such as ours. Hence for our experiments, we report precision, recall and F1-score. The results are evaluated on a held-out dataset of $\sim 3500$ users after a 80,10,10 train/test/validation split of the dataset. Furthermore, the models were trained for 10 epochs (in all experiments) with *repeat* mode to prevent data exhaustion.

## 5.1 Synthetic Masking of Labels

To evaluate the robustness of our model under increasingly sparse supervision, we design a synthetic masking experiment that artificially hides a fraction of the already limited labeled responses in the data. It is important to note that the underlying dataset is inherently sparse due to survey routing logic; most questions are not shown to each user, and the observed labels comprise only a small subset of the total response space. In this experiment, we apply additional random masking on top of that existing sparsity to simulate even more aggressive missingness.

Specifically, for each user vector $x$, we select a random subset where $m_i = 1$ i.e., where a question was shown and the response is known; and set those values to zero. This masking is performed at rates of 15%, 30%, and 50%, representing increasingly constrained test-time observation. The no-mask setting corresponds to evaluation on the original dataset, without synthetic masking, but still reflects the natural sparsity of fielded responses. Table 1 summarizes these findings.

Table 1: Performance metrics under varying levels of synthetic masking applied to observed labels during evaluation. Note that these masking rates are applied on top of the already sparse label space due to survey routing logic.

| Setting (% of Masking) | *Precision* | *Recall* | *F1* |
|---|---|---|---|
| 50 | 0.9792 | 0.2576 | 0.4079 |
| 30 | 0.9573 | 0.6209 | 0.7532 |
| 15 | 0.9279 | 0.7448 | 0.8263 |
| 0 (No artificial mask) | 0.8859 | 0.8121 | 0.8474 |

As the masking percentage increases, we observe a consistent rise in precision and a corresponding drop in recall and F1. This trend reveals the model's increasing conservatism under high uncertainty. It becomes less willing to make positive predictions, and hence makes fewer mistakes, but also misses more true positives. At 50% masking, recall deteriorates sharply, lowering the F1 score despite very high precision.

This pattern is an expected and informative outcome. It indicates that the model maintains high confidence in its predictions when label availability is abundant but degrades gracefully as available supervision decreases. In practical terms, this result demonstrates robustness under conditions where different surveys vary in depth and routing logic. It also reinforces the challenge of imputing long-tail behaviors with little to no supervision.

Table 2: Ablation results for two core model components: transformer and mask-aware supervision.

| Ablation Setting | *Precision* | *Recall* | *F1* |
|---|---|---|---|
| I[a] + M[b] | 0.8832 | 0.7706 | 0.8231 |
| I + T[c] | 0.8308 | 0.6962 | 0.7575 |
| I + T + M | **0.8859** | **0.8121** | **0.8474** |

[a]I defines the base model
[b]M represents masked supervision
[c]T represents transformer block

## 5.2 Ablation: Self-Attention & Masked Supervision

To evaluate the contribution of column-wise attention to imputation performance, we conduct an ablation in which the transformer block is removed from the architecture. Instead of computing contextualized representations via feature interactions, each embedded response key is passed through a shared MLP in isolation, without attending to other responses. This setup eliminates the model's ability to model co-occurrence patterns or structural dependencies across different survey responses. It serves as a test of whether the architecture benefits from learning latent inter-feature semantics. We evaluated this variant using standard imputation metrics under a fixed thresholding regime (0.5 cutoff), and compared it against the full model. Findings are presented in Table 2.

The ablated model achieves reasonably high precision, but recall drops notably compared to the full model, leading to a lower overall F1 score. This indicates that without column-wise attention, the model becomes more conservative. It is still able to make correct positive predictions, but is unable to recover as many true positives. In effect, the model lacks the context needed to confidently activate less obvious or indirectly related behaviors. This result confirms that column-wise attention is a meaningful contributor to performance in this setting. By enabling the model to learn associations between otherwise distant or structurally unrelated response keys, attention allows for richer imputations, particularly in the presence of high-dimensional sparsity.

To understand the role of mask-aware supervision in guiding model learning, we ablate this mechanism by training the model on all entries in the input vector $x$, regardless of whether a label was observed. In this variant, the loss is computed across the entire response space, treating all unasked questions (where $m_i = 0$) as having meaningful supervision targets. This setting deviates from the logic-aware approach used in the full model, where supervision is restricted only to entries that were explicitly shown to the user during the survey. Although the ablated variant leverages more data points during training in a naive sense, it introduces noise and label ambiguity by treating structurally missing entries as valid targets.

This variant suffers the most pronounced performance drop among the ablations. Both precision and recall degrade, and F1 is substantially decreased. These results confirm that ignoring the supervision mask during training injects noise into the learning process. The model attempts to predict values for entries with unknown ground truth, effectively confusing absent data. The lowered precision and recall suggest that this leads to overfitting on unreliable targets and under performance on valid ones. This ablation highlights the importance of aligning the learning objective with the known structure of the data collection process. Masked supervision not only respects the logic of survey fielding, but also acts as a safeguard against spurious correlations and label noise.

## 6 Responsible Imputations

In high-impact industrial applications, particularly those involving user behavior modeling for segmentation and personalization, the integrity of model predictions must be examined not only through the lens of performance metrics but also through fairness and representational equity. To that end, we conducted a dedicated fairness audit of our imputation model grounded in two distinct but complementary perspectives of responsibility: fit-based responsibility and outcome-based responsibility. Table 3 summarizes the sample sizes for each subgroup included in the fairness audit. The analysis was conducted on a subset of approximately ∼40,000 users.

Table 3: Sample sizes of each subgroup used in the fairness audit. These statistics provide context for interpreting group-level performance and error metrics.

| Group | Sub-Group | Counts |
|---|---|---|
| Age Group | 18-24 | 658 |
| | 25-34 | 1186 |
| | 35-44 | 1542 |
| | 45-54 | 1447 |
| | 55-64 | 1337 |
| | 65+ | 1254 |
| Gender | Male | 3404 |
| | Female | 4020 |
| Ethnicity | Asian | 466 |
| | Black | 1247 |
| | Hispanic | 912 |
| | Middle Eastern/ North African | 52 |
| | Native American | 261 |
| | White | 5270 |
| | Other | 120 |

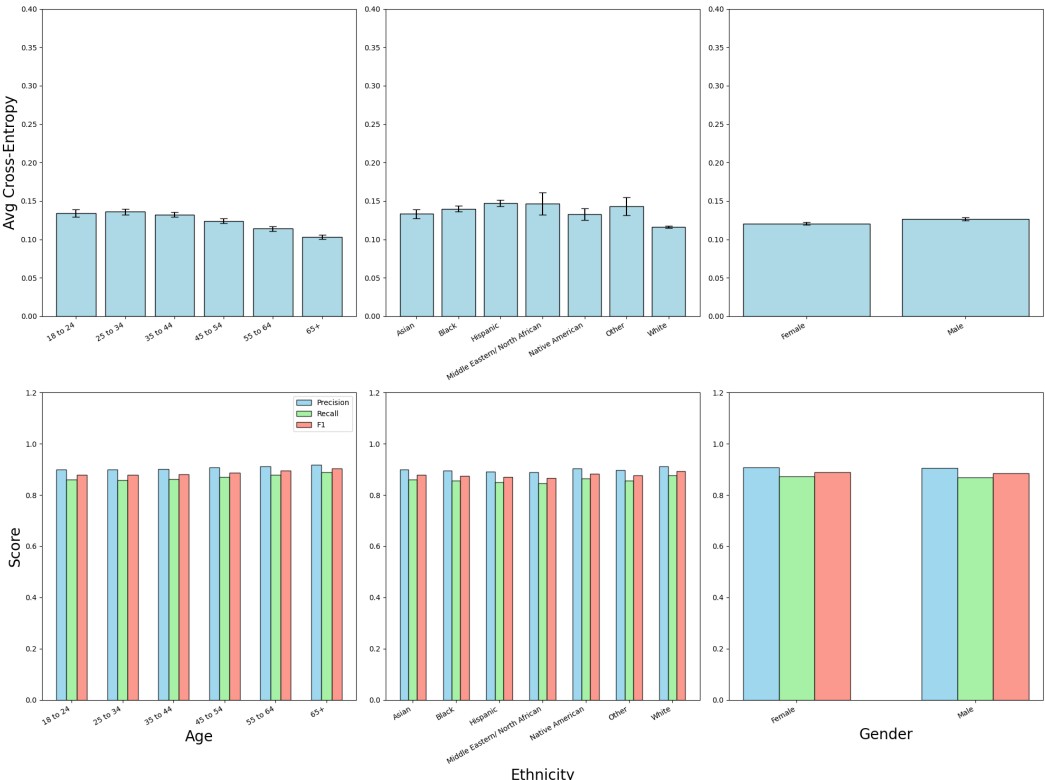

Figure 3: Fairness audit. **Top**: average cross-entropy loss with error bars for age buckets, ethnicity categories, and gender. **Bottom**: group-wise precision, recall, and F1 scores. The audit captures both fit-based responsibility (via cross-entropy) and outcome-based responsibility (via classification metrics).

Fit-based responsibility emphasizes that the model should demonstrate equitable learning behavior across subgroups. That is, its ability to fit observed data, to correctly learn from and reconstruct user responses, should not vary substantially depending on user identity attributes such as gender, age, or ethnicity. We operationalize this by computing the average cross-entropy loss for each group, along with its margin of error. This analysis, shown in Fig 3 revealed minor disparities. Younger users exhibited slightly higher cross-entropy values compared to older users, and some ethnicity subgroups had marginally elevated error, particularly those with smaller sample sizes. However, none of these differences crossed thresholds that would suggest structural unfairness or bias. The margins of uncertainty ($\pm 0.005$) around these values further confirmed that observed variations were within the bounds of sampling noise rather than indicative of model prejudice.

In parallel, we conducted an outcome-based responsibility analysis, which evaluates how well-calibrated the model is across demographic lines. Here, the focus shifts from learning effectiveness to the expected quality of outcomes. We quantified this using precision, recall, and F1 scores disaggregated across these subgroups. The results indicate consistently high performance across the board. For instance, gender-wise comparisons showed near-identical F1 scores for male and female users, with both groups exhibiting strong and balanced precision and recall. A similar pattern held across age bands, where older users achieved marginally higher F1 scores, potentially reflecting more stable or habitual response patterns. Ethnicity-based analysis revealed no group falling below parity in imputation quality, with F1 scores across groups ranging within a narrow and acceptable band.

Together, these two lenses provide a multidimensional view of fairness. While fit-based metrics confirm that the model learns equally well from all segments of the population, outcome-based measures assess whether the model's predictions carry different levels of confidence or reliability across those same segments. Integrating both into our audit reflects a comprehensive and responsible approach to fairness in model evaluation, one that goes beyond surface-level parity to probe the deeper mechanics of equity in learning and inference. Our fairness analysis shows no evidence of systematic disadvantage for any group. The model appears to generalize equitably, and any minor disparities observed are well within acceptable statistical variance. These findings reinforce the viability of deploying the model in production settings while also underscoring the importance of continual fairness monitoring, especially as the model encounters new populations or adapts to evolving data distributions over time.

## 7   Conclusion and Future Work

In this work, we presented a logic-aware imputation framework tailored for structured survey response data; a uniquely sparse and high-dimensional domain ubiquitous in behavioral modeling for industrial applications such as AdTech, personalization, and consumer intelligence. By framing the imputation problem as a supervised task over masked binary labels and treating user responses as unordered, multi-hot vectors, we designed a Transformer-based model architecture that respects the structural and operational realities of survey data collection.

The proposed model introduces multiple innovations in the field to handle these challenges: column-wise attention for learning inter-response dependencies, mask-aware supervision that aligns with routing logic, Gaussian noise injection for robustness, and a causality layer that enforces logical and business constraints post-inference. Through experimentation, we demonstrated the model's effectiveness under synthetic masking regimes, outperforming strong ablations and naive baselines. Our responsible audit further validated the approach along both fit-based and outcome-based axes, revealing equitable performance across key groups.

While these results establish a strong foundation, several promising directions remain. The current model does not explicitly capture causal dependencies between responses; incorporating structural priors or causal graph constraints could improve coherence and reduce over-imputation. Future work could explore differentiable constraint learning to internalize consistency within the model itself. Finally, our framework could be extended to semi-supervised or multitask settings, where imputed outputs directly inform downstream classifiers or audience definitions. Altogether, this work lays the groundwork for responsible, scalable, and interpretable survey imputations enabling better behavioral understanding while maintaining fairness and logical integrity.

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

## A  Experiments & Implementation Details

Model is implemented in TensorFlow and trained using the Keras functional API. The model consists of standard layers - embeddings, dense, multi-head attention, and a prediction head with dynamic masking.

The implementation details are given below.

- Model Size: $\sim 355$ million parameters
- Optimizer : Adam with default settings of $\beta_1 = 0.9$, $\beta_2 = 0.99$, and learning rate of $1e-03$.
- Loss: Binary Cross Entropy
- Hyperparameters : Included parameters like activation function for dense layers, gaussian noise ($G_1$ & $G_2$) and attention heads. The choices were carefully determined during research, and hyperopt tuning library was used to find the optimal values for training.

- Batch Size: 128

- Epochs: Trained to 30 epochs with *training_steps_per_epoch* = 1000 and *validation_steps_per_epoch* = 100. The data was loaded with *repeat* enabled to prevent data exhaustion; *shuffle* was turned on to discourage any ordering.

- Hardware: Model was trained for 4 hours on a single A10G instance GPU using AWS Cloud Infrastructure.

For each of the experiments, we used the same hardware configuration as used during training. Each experiment was trained to epochs in $\sim 1$ hour. The instance we used lists the following configuration.

- Memory : 128 GB

- Storage : 900 GB

- GPU Memory : 24 GB

- vCPUs : 32

