# OpenReview forum: "Responsible Imputation of User Behavior Surveys via Mask-Aware Transformers"
_NeurIPS.cc/2025/Workshop/Reliable_ML — NeurIPS 2025 - Reliable ML Workshop_

### Official Review · Reviewer_Ed83 · 2025-09-10
**Clear Approach to Survey Imputation, but Limited Novelty**

**Rating:** 7
**Confidence:** 2

**Review:**

**Review**

This paper addresses the problem of imputing missing values in survey data, where structured missingness arises due to routing logic and platform constraints. The authors propose a Transformer-based model with mask-aware supervision, column-wise attention, noise injection, and a logical post-processing step. They also include a fairness audit across demographic subgroups.

**Strengths**

The paper is clearly written and well organized, with nice plots and figures, making the method and motivation easy to follow.
It is refreshing to see attention given to survey data imputation, which is less frequently studied compared to general tabular imputation.

**Weaknesses**

The technical novelty is limited: the main contribution is an application of standard Transformer architectures with some domain-specific masking and simple post-processing rules.
The proposed method is relatively straightforward from an engineering standpoint, and the originality is modest.
A major gap is the lack of quantitative comparisons to existing imputation baselines (e.g., MICE, MissForest, GAIN, SAITS, etc.). Without such benchmarks, it is hard to assess the true benefit of the approach.
Evaluation is limited to a proprietary dataset, so the generality of the results remains unclear.

**Suggestions**

Strengthen the experimental section with baseline comparisons on both proprietary and (if possible) public datasets.
Clarify whether the logical enforcement could be integrated into the model itself rather than only as post-processing.

**Overall**

While the contributions are incremental, the paper is very well written and brings useful attention to a practical but underexplored problem domain. For a workshop, this makes for an interesting applied case study, but stronger baselines and a clearer claim of novelty would be needed for a main conference submission.

---

### Official Review · Reviewer_x5iE · 2025-09-19

**Rating:** 6
**Confidence:** 4

**Review:**

## Summary
This paper considers the scenario where the datasets are survey results, which have an inherent data structure, including logic routing (for assigning survey questions), platform constraints, and user fatigue results. In this scenario, the aim of this work is to design a model for imputing the choice of missing responses. The challenge here is that the missing is non-random and logic-based (artificially designed and targeting specific users). To address the issues, the paper proposed an encoding scheme honoring the

First, I have to admit that I am not an expert in this direction, so my evaluation might be biased and inexact.

## Strengths and Weaknesses
### Strength
I like the presentation of this paper, which is very clear and easy to follow. The proposed method is sound and directly targets the inherent challenge of this problem (e.g., the non-randomness and logic-based nature of missing responses).

### Weakness
1. I think the considered problem, though interesting, is very niche, specifically targeted to multiple-choice surveys.
2. The novelty is limited and straightforward. It is not necessarily a bad thing, as I actually prefer the simplicity and the natural soundness of the method, but I think that it would limit the contribution of this draft. Since the problem considered is niche, I would love to see some creativity in the approach. But again, I am not an expert in this survey direction, so my evaluation might be biased and unfair.
3. The experiments were conducted on a private dataset, and the dataset description was minimal, which makes it hard to validate the trustworthiness of the experiments. But I totally understand the privacy concerns of this task, so I would treat this lightly.


## Final comments
Overall, I think this makes a good workshop paper. The problem setting is clear, and the proposed method sounds good. There are some concerns about the setting of experiments, which was described very sketchily and evaluated on a private dataset. But for a workshop paper, I think it is acceptable.